# On the History of Ecosystem Dynamical Modeling: The Rise and Promises of Qualitative Models

**DOI:** 10.3390/e25111526

**Published:** 2023-11-08

**Authors:** Maximilien Cosme, Colin Thomas, Cédric Gaucherel

**Affiliations:** 1UMR AMAP, INRAE, University of Montpellier (Faculté des Sciences), IRD, CIRAD, CNRS, 34398 Montpellier, France; 2UMR DECOD, Institut Agro Rennes-Angers (Campus Rennes), 65 rue de Saint-Brieuc, 35042 Rennes, France; 3IBISC, University of Evry, 91025 Evry, Francecedric.gaucherel@inrae.fr (C.G.)

**Keywords:** qualitative models, social ecological modeling, discrete-event models, history of modeling, state transition graphs

## Abstract

Ecosystem modeling is a complex and multidisciplinary modeling problem which emerged in the 1950s. It takes advantage of the computational turn in sciences to better understand anthropogenic impacts and improve ecosystem management. For that purpose, ecosystem simulation models based on difference or differential equations were built. These models were relevant for studying dynamical phenomena and still are. However, they face important limitations in data-poor situations. As a response, several formal and non-formal qualitative dynamical modeling approaches were independently developed to overcome some limitations of the existing methods. Qualitative approaches allow studying qualitative dynamics as relevant abstractions of those provided by quantitative models (e.g., response to press perturbations). Each modeling framework can be viewed as a different assemblage of properties (e.g., determinism, stochasticity or synchronous update of variable values) designed to satisfy some scientific objectives. Based on four stated objectives commonly found in complex environmental sciences ((1) grasping qualitative dynamics, (2) making as few assumptions as possible about parameter values, (3) being explanatory and (4) being predictive), our objectives were guided by the wish to model complex and multidisciplinary issues commonly found in ecosystem modeling. We then discussed the relevance of existing modeling approaches and proposed the ecological discrete-event networks (EDEN) modeling framework for this purpose. The EDEN models propose a qualitative, discrete-event, partially synchronous and possibilistic view of ecosystem dynamics. We discussed each of these properties through ecological examples and existing analysis techniques for such models and showed how relevant they are for environmental science studies.

## 1. Introduction

Ecosystem ecology emerged in the 1950s–1960s in the context of the growing popular and political awareness of radioactive pollution and land degradation [1]. New methodologies appeared during this period, for instance the use of radioactive tracers to study material flows. The 1950s–1960s also correspond to the computational turn [2], during which computer simulations and analysis of complex models became possible. The ability to understand ecosystems with integrated equation systems became possible at that time, as more variables and processes connecting them could be taken into account. Computers allowed growing early models into more realistic numerical representations and dynamics. Ecosystem models were introduced for simulation and systems analysis on a multidisciplinary basis [3], as any social-ecological system involves a wide range of domains. Most of them were quantitative and were mostly used for designing appropriate management policies (e.g., [4,5]). However, these quantitative models required precise information on the shape and intensity (i.e., parameters) of ecological interactions, which are often costly to obtain, thus making calibration challenging [6]. Moreover, they are often unable to incorporate expert knowledge and direct observations, which are a frequent source of qualitative information in social-ecological systems [7]. For example, a concrete study in dendrochronology requires a detailed and complicated calibration, still subject to equifinality (similar outputs for distinct parameter sets) and instability [8]. Consequently, alternative avenues such as qualitative models [9] were soon investigated.

Qualitative ecological models emerged as a complementary approach to quantitative models [10], aiming to describe coarse-grained ecosystem dynamics. Qualitative models may be appropriate when quantitative information on parameters and variables are insufficient and/or when the modeling objectives do not require a quantitative response. As qualitative model outputs are relatively independent from parameter values, their results are thus less precise yet more general [11]. Qualitative models accept broader inputs, and thus they can make use of both expert and scientific knowledge, resulting in more robust outputs. Since the 1980s, several qualitative frameworks have been developed, such as loop analysis [9], qualitative reasoning models [12], fuzzy models [13] or discrete-event models [14,15]. Qualitative models aim to address several issues, such as which information is required to disambiguate model predictions [16] or how a system can achieve qualitative stability [17].

Models are described both by their phenomena of interest and by their mathematical formalization, the choice of the latter often resulting from the former. For instance, the nitrogen cycle is generally studied quantitatively and modeled by a biogeochemical model expressed as a set of differential equations. Alternatively, assessing the (qualitative) response sign (+, −, 0) of a resident species to an invasion can be achieved through qualitative methods such as loop analysis. Mathematical properties can be combined in a multitude of ways in order to design a model, resulting in different perspectives on system dynamics. The modeling process thus includes preliminarily choosing the best combination of properties for the desired objectives.

In Section 2 of this article, we provide a non-exhaustive historical overview of dynamical ecosystem models, with an emphasis on qualitative models, and discuss their respective merits and limitations. Then, in Section 3, we isolate the most salient model properties and discuss their relevance for (1) grasping (at least qualitative) system dynamics while (2) keeping the model parsimonious, (3) explanatory and (4) predictive. In Section 4, we propose the ecological discrete-event network (EDEN) framework for modeling ecosystem dynamics while meeting simultaneously these objectives. While other qualitative approaches sometimes address some of these objectives separately, using the discrete-event and asynchronous update mode in EDEN allows for providing multi-objective yet simple qualitative models. Finally, we discuss the general relevance of qualitative modeling approaches for studying ecosystems.

## 2. Overview of Ecosystem Ecology and Dynamical Modeling Approaches

The term “ecosystem ecology” was coined in the 1950s, continuing major modeling breakthroughs in life sciences in the early XXth century [18]. This sub-field of ecology predominantly originated in the USA through the work—among others—of E.P. Odum [19]. Inspired by dynamical systems theory [20], ecosystem ecology mostly aimed to understand fluxes of radioactive material in ecosystems using concepts and techniques from early biogeochemistry [21,22]. Tracing the radionuclides from their emission sources to each compartment (e.g., plants, rivers or soil) helped to reveal material flows in ecosystems [23].

Although dynamical modeling was already well established in ecology, it was mainly used in population ecology to model the growth of single [24,25,26,27,28] and interacting populations [29,30]. However, ecosystem ecology benefited from the development of computational modeling [1,31], which enabled the simulation of the dynamics of complex and larger ecosystems.

### 2.1. Compartment Modeling

A compartment model [3,32] generally consists of (1) variables representing compartments, (2) equations representing flows between compartments and (3) parameters representing the intensity of these flows [33] (Figure 1). These models are mostly formalized as ordinary differential or difference equations (e.g., [34]). To account for uncertainty, stochastic parameters or events can also be included (e.g., [35,36,37]).

Compartment models were at the core of systems ecology, which was the meeting point between ecosystem ecology and simulation models (Figure 1). Systems ecology was strongly underpinned by a holistic philosophy [38,39]. Following Clements’ ideas from the early XXth century [40], it emphasizes the irreducible complexity and the self-regulating nature of ecosystems and emerges from their internal relations [41,42]. Drawing upon emerging simulation techniques, systems ecologists proposed the “total-system modeling” approach which aimed to include “abiotic, producer, consumer, decomposer and nutrient subsystems [in order to] assure that the modeling effort [plays] the integrative role delegated to it” ([43], p. 2). This approach was central to the Biome studies investigating the response of biomass to changes in external conditions [44]. Today, this integrative approach is still highly promoted in ecosystem modeling, although with different formalisms such as individual-based models (e.g., [45,46]). These models allowed researchers to predict the nutrient–productivity relationships in macrophyte populations [47] or the impact of habitat fragmentation on larger animals [48]. However, as pointed by Jørgensen [49], technical limitations greatly limit data collection and thus the estimation of precise parameter values that are required for making reliable predictions. Additionally, while parameters generally remain constant during simulations, real systems actually display variable parameter values (i.e., biological plasticity and those where Jørgensen [49] calls for structural dynamics). In addition, such multiple, non-linear differential equations often exhibit sensitive dependence to initial conditions and to multiple feedback, thus leading to so-called chaotic behaviors.

**Figure 1 entropy-25-01526-f001:**
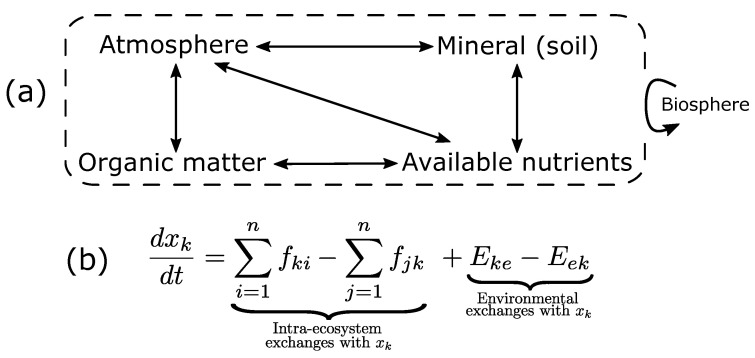
**Compartment model of a terrestrial ecosystem.** (**a**): Ecosystem compartments linked through matter and energy fluxes represented by arrows. Each compartment may be split into several sub-compartments (e.g., organic matter can be split into living and dead organic matter). The dotted box represents ecosystem boundaries, through which the ecosystem is also linked to the whole biosphere through biotic and abiotic processes. Adapted from [50]. (**b**): A system of ordinary differential equations (ODEs) of *n* variables ${\left(x_{k}\right)}_{k\leq n}$ for k∈{1,…,n} and $n^{2}+2n$ parameters. With $f_{ki}$ and $f_{jk}$ describing incoming (from *i* to *k*) and outgoing (from *k* to *j*) fluxes within the ecosystem, respectively, and $E_{ke}$ and $E_{ek}$ describing incoming and outgoing environmental exchanges, respectively [51].

One way to cope with such structural dynamics is the application of thermodynamic principles to ecology [52]. This holistic approach was developed in the 1950s–1980s and proposed general predictions about ecosystem development [53,54]. This directional development is driven by goal functions (or orientors), which tend to be maximized or minimized over time [55]. These include emergy (i.e., content of solar energy equivalent in matter [56]) and exergy (i.e., a system’s deviation from thermodynamical equilibrium [57]). Self-organization processes such as ecosystem development have been explained by goal function maximization [54]. Such studies consists in identifying measurable state variables that may follow extremal principles supposedly justifying the system behavior. The first thermodynamical principles in ecology were proposed in the 1920s [58], and most of their theoretical foundations come from the thermodynamics of open systems (i.e., systems exchanging matter and energy with their environment) far from equilibrium [52,59]. While these ideas have undergone numerous developments and unification since the 1970s [60], they remain poorly used for applied issues [61].

### 2.2. Qualitative Modeling Approaches

Qualitative models rely on the abstraction of quantitative phenomena by focusing on qualitative changes in system state or in the sign of variables’ derivatives. This abstraction generally aims to account for the lack of precise measurements and/or to propose a complementary way of interpreting phenomena. In contrast to quantitative models, they do not necessarily rely on numeric inputs and mostly use relational (e.g., >, < or =), set (e.g., inclusion or exclusion) or logical (and, or, not) operations. Despite their recent rise with the computational turn in ecological modeling, their history goes back to the late XIXth century, with the work of Lorenzo Camerano.

#### 2.2.1. Camerano’s “Reaction Networks”

Lorenzo Camerano is an Italian entomologist of the XIXth century, mostly known for his seminal representation of an ecological community as an interaction network [62]. The dynamical model provided alongside this network has been overlooked, although it is probably one of the first dynamical trophic models in the history of ecology [63]. This qualitative model of trophic cascades consists of variables representing species (or functional groups) and interactions between these variables (noted by an arrow →). For example, if a species *A* feeds upon a species *B* and *B* feeds upon *C*, we have:A→B=B−andC+
where = states the result of the interaction and − (resp. +) a negative (resp. positive) effect on a population growth rate. The decrease in *B* is instantaneously associated with an increase in *C* (i.e., its resource).

This model is qualitative because it focuses on the increase or decrease in a population’s growth rate without quantifying it. Ecological interactions are split into separate rules, which also likely makes this model the first rule-based model in ecology. Since any cause is followed by its immediate and non-random effects, this model is deterministic (i.e., the future is fully predictable given an initial condition). This is further confirmed by the figures depicting non-branching trajectories and the main text (“a reaction which […] bring[s] about certain determinate changes”). Camerano informally generalizes cascading effects using a deterministic mechanical analogy based on sound vibrations. Unfortunately, the work of Camerano was quickly forgotten. It was rediscovered one century later [62] and had no known impact on ecology. However it highlights the early need by some naturalists for qualitative models as a theory-building tool.

#### 2.2.2. Loop Analysis

Almost one century after Camerano’s model, Levins [9] proposed a qualitative analysis of ecological interaction graphs (Figure 2a), called loop analysis. It aims to predict the consequences of press perturbations [64] in an ecological system. Levins proposed loop analysis mainly as complementary to total-system models [10], the latter promoting precision and realism at the cost of generality [11]. This technique emerged in economics in the 1920s [65] but has been extended to address ecological issues [16,66].

Loop analysis is applied to signed directed graphs (digraphs). In these graphs, nodes represent ecosystem components (generally species), and edges represent interactions. Additionally, edges are directed and have a positive (symbolized by →) or negative (symbolized by ⊸) sign representing the effect of interactions. These graphs are built from the community matrix (representing the constant interaction signs) derived from the Jacobian matrix of a system of differential equations near equilibrium (Figure 2b). Loop analysis assesses how the effects of an external press perturbation (affecting growth rates positively or negatively) are mediated through an ecological network. In particular, it aims to predict whether a change in the growth rate of a variable has a positive (+), negative (−) or neutral effect (0) on the equilibrium abundance of each variable [67] (Figure 2c). More detail on calculations can be found in [68]. Loop analysis relies on a particular epistemology seeking to deduce rigorous predictions in spite of limited information on parameter values and response forms [69]. The resulting predictions are thus robust to quantitative variations in parameter values in simple systems.

**Figure 2 entropy-25-01526-f002:**
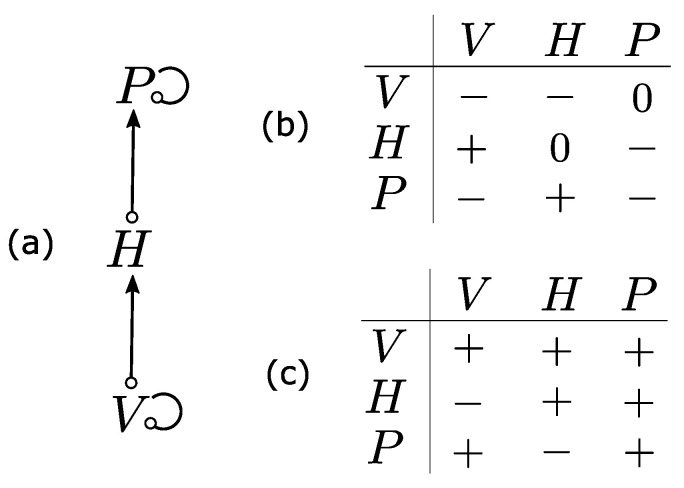
**Loop analysis of a signed digraph**. (**a**) A signed directed graph (digraph) representing the interactions of the vegetation–hare–predator interaction network. Reciprocal interactions are merged into a single bidirectional edge. (**b**) Community matrix whose elements are the sign of the corresponding pairwise interaction represented in (**a**). (**c**) Effect of an increased growth rate of each variable on others variables. Adapted from [70,71].

A well-known property of signed digraphs is ambiguity (i.e., the indeterminate response of a variable to a press perturbation), which can only be resolved by considering the parameter values. This property helps ecologists by pinpointing the additional data required to disambiguate the model predictions [69,72]. Ambiguity can be circumvented by keeping the network structure constant and randomly drawing interaction parameters and running each resulting quantitative model independently. This so-called ensemble ecosystem modeling procedure generates a set of ecosystem models whose outputs are then filtered and analyzed [73].

Using loop analysis, [74] identified keystone species in marine ecosystems and [75] proposed management actions to preserve kelp populations. However, the fact that the applicability of loop analysis is limited to the local neighborhood of a point equilibrium has at least two consequences. First, the system has to have a reachable equilibrium, which is not guaranteed. Second, the sign of interactions can only be constant, thus restricting the analysis to linear and monotonic interactions [69], although non-monotonic effects are known to play an important role in ecosystem dynamics and stability [76].

#### 2.2.3. Qualitative Reasoning

Qualitative reasoning emerged in the 1970s and is an area of artificial intelligence designed to model qualitatively the continuous behavior of a system. It emerged in the 1970s as a means to model physical systems based on qualitative information about system interactions and variables. Predictions about system dynamics are thus possible even when information is scarce or non-numerical. Qualitative reasoning has been used in ecology since the 1990s [77] and since the 2000s for ecosystem issues [78]. Contrary to loop analysis, these models are generally associated with a simulation engine, with QSIM [79] and Garp3 [80] being the most used in ecology. QSIM simulates a system of qualitative differential equations (i.e., qualitative abstractions of ordinary differential equations (ODEs)). On the other hand, Garp3 aims to represent expert knowledge by building model fragments and assembling them to compute the whole system dynamics. Like signed digraphs used in loop analysis, these models are based on a set of qualitative variables, whose interactions are signed, unweighted and non-probabilistic. Each variable is described by its magnitude (e.g., zero, low, normal, maximum) and its direction (i.e., increasing, stable or decreasing). As in loop analysis, ambiguity (i.e., indeterminacy of change) may occur. Every ambiguous situation leads to alternative trajectories, one for each possible resolution of the ambiguity. The resulting dynamics are thus non-deterministic.

Qualitative reasoning has been used to model Brazilian savanna (cerrado) dynamics [78], avian communities’ changes in response to farming practices [81] and the impact of Paleozoic land plant evolution on the carbon cycle [82]. QSIM guarantees that each simulated trajectory will be observed in the corresponding ODE [79]. However, the main limitation to qualitative simulations is that they may produce spurious behaviors, that is, trajectories predicted by the qualitative model but not predicted by any corresponding ODE.

#### 2.2.4. Models Based on States and Transitions

Since the early days of ecology, dynamics have been represented as sequences of discrete states and transitions, such as vegetation succession diagrams (e.g., [40,83,84]). These diagrams are conceptual models representing temporal changes in vegetation composition as a directed graph [85]. These changes in vegetation state are generally considered reversible but can also include irreversible transitions (e.g., [83] showing that bad management induces “more or less permanent deterioration of the soil”, inducing irreversible vegetation transitions). These diagrams have largely been used for illustrating the Clementsian succession theory, which states that vegetation, if left undisturbed for a sufficiently long time, develops predictably through vegetation phases towards a fixed state called climax [40]. These concepts were the basis for rangeland management from the 1910s to the 1990s [86,87,88]. However, some rangeland ecologists were questioning traditional ideas about vegetation succession, asserting that rangelands can undergo irreversible transitions preventing the return to a climax vegetation, thus contradicting the Clementsian theory. In this context, [89] developed a non-formal modeling approach called state-and-transition models (STMs) to account for (i) the existence of multiple pathways between vegetation states and (ii) irreversible transitions in rangeland states, and (iii) to gather expert knowledge and serve as a tool for decision-making [90].

Like succession diagrams, STMs are directed graphs whose nodes and edges represent ecosystem states and transitions, respectively (Figure 3), which are both derived from direct observations. They are generally used as dynamical databases for rangeland management, not based on time but on events. Note that similar graphs representing states and transitions were anecdotally used in the 1970s under the term “behavior graph” [3] or “replacement sequence” [91] where, in the latter, transitions have specific time durations. While succession diagrams and STMs are not formal models, several authors have proposed using Markov chains for modeling vegetation succession [92,93] and rangeland dynamics [94].

Central STM concepts have been well defined by [90]: (1) A state is “a recognizable, resistant and resilient complex of two components, the soil base and the vegetation structure”; (2) A transition is “a trajectory of system change away from the current stable state that is triggered by natural events, management actions, or both”; (3) Reversible transitions occur within states, which may consist in several community phases (e.g., reversible changes in plant species composition); (4) Irreversible transitions (at least, in time scales relevant to management) occur between states.

As in qualitative reasoning models, any state in a STM may have several outgoing non-probabilistic transitions, i.e., they represent all possible states and transitions. They are generally built upon observations and thus cannot predict novel ecosystem structures [96,97]. This limitation led scientists to combine them with mathematical tools such as dynamical Bayesian networks [98] or Markov chains [94].

Each modeling framework emphasizes new aspects of a system and thus widens the range of questions that can be addressed. Each of them simultaneously imposes methodological and practical limitations. On the one hand, methodological limitations over-determine both the systems or phenomena that can be modeled. They originate from the methodology or the mathematical tools employed to model the system. For instance, STMs cannot be used to predict unobserved transitions or loop analysis cannot be applied to non-equilibrium systems. On the other hand, practical limitations generally stem from the relationships between the data and the model (e.g., limiting data availability). For instance, insufficient data may impede quantitative and robust parameterization of compartment models. In ecosystem models such as the dendrochronological study previously cited, the fine tuning of the tree growth module clearly shows limitations in terms of calibration, of computation time required, of uncertainty and, ultimately, of choices in representing the phenomenon [8]. A qualitative model would have been less precise, but much more accurate, in computing the effects of the driving factors. The choice of model properties can contribute to circumvent some practical limitations. For instance, stochasticity or qualitative variables are a privileged way to cope with the lack of data. Models can thus be viewed as assemblages of properties specifically designed to address specific questions. Therefore, understanding the limitations of each model property would strongly help modelers in their choice between alternative methods (or help them to design new ones) to better tackle (ecological) questions.

## 3. Ecosystem Models’ Properties and Their Limitations

Dynamical ecosystem models seek to predict and/or explain ecosystem trajectories, either by reproducing some of their internal interactions and external influences (process-based approach, as in most compartmental models), or by reproducing portions of the observed phenomena (phenomenological approach, as in state-and-transition models). Depending on their objectives, these models improve our understanding and/or provide solutions to applied issues. These objectives are generally reflected in their mathematical properties. For instance, loop analysis aims to assess the direction of change of an equilibrium population size to a press perturbation and thus only focuses on the interaction signs. Therefore, the formalism and properties used to design a model often result (at least partly) from a motivated choice.

We now summarize our present scientific objectives. Our modeling framework is aimed at:1.Grasping the qualitative dynamics of the system (i.e., not requiring any quantitative values, based on state and transitions or on state variable variations);2.Making as few assumptions as possible about interactions (on parameters and/or functional form);3.Being explanatory in a general sense, i.e., answering to why-questions [99]; and4.Being predictive, i.e., forecasting the future state of a system before the system reaches it [100].

Based on these objectives, we will discuss the benefits and limitations of general model properties. Therefore, this should not be seen as a critique *per se* of each property, but only a study with respect to the aforementioned objectives. Further (Section 4), we will introduce the EDEN framework and show how it helps circumventing such limitations by combining qualitative and discrete-event properties with an asynchronous update mode.

### 3.1. Quantitative or Qualitative Variables?

Quantitative or semi-quantitative models such as compartmental models generally involve differential/difference equations. They provide precise and unambiguous predictions that can (in principle) be readily measured [66]. One strategy is to “sacrifice generality” [67] and greatly detail interactions in the model. However, information about processes is often partial and thus ecological parameter estimations remain limited [6,66]. This issue is generally addressed by a strategy of simplification, reducing the model size (i.e., the number of variables and parameters) in order to reduce the number of required measurements (the “sacrificing realism” strategy [67]).

Reducing the model size is only one of the possible simplifications. An alternative strategy to the previously mentioned ones is to simplify the variables’ behavior through a *qualitative approach*. Instead of focusing on their precise values, the emphasis can be put on crisp (e.g., qualitative reasoning) or overlapping (e.g., fuzzy models) intervals or on their signs of variation (as in Camerano model or loop analysis). In this way, qualitative approaches provide a complementary way to see system dynamics by forcing a focus on drastic changes in the studied system. Some other attempts in systems biology such as auto-catalytic networks and studies in chemical organization theory display clear complementarities with the EDEN framework [101]. Such approaches in systems biology focus on the co-existence of species in a similar and often qualitative way.

Moreover, much ecological knowledge is qualitative [102] and is held by ecosystem stakeholders [103]. Taking into account this qualitative information makes model building easier and provides general results [11]. Also, some ecological questions do not necessarily require quantitative information, such as the identification of keystone species [74].

Therefore, objectives (1) and (2) suggest adopting a qualitative approach. Note, however, that the predictive power of some qualitative models (such as loop analysis) generally decreases as the number of variables increases due to higher indeterminacy in model responses [16], as these models become useless with more than ten or twelve variables. This will be addressed in Section 4.2.

### 3.2. How Changes Occur over Time: The Variables Update Mode

In most dynamical models, the changes in variable values at a time *t* depend on the system state at time t−1. Hence, at each time step, all possible changes are executed simultaneously, i.e., all variables are updated synchronously. Such a synchronous update mode is appropriate in quantitative models but becomes problematic when variable values represent intervals such as “low”, “medium”, and “high”. This implicitly assumes that all qualitative changes (e.g., shifts from “low” to “medium”) are equally fast, which is a strong assumption for social, biological and ecological processes. These major changes happen at regular time steps, as if the same “global clock” was driving the behavior of each ecosystem component.

Conversely, it has been argued that ecosystems are distributed [41]. It means that each ecosystem component does not respond in a synchronized way with others but rather responds at its own speed. This implies that responses to a given stimulus may be delayed between components due to their mutual independence or to different reactivity. Synchronous qualitative dynamical models may fail to predict the qualitative behavior of a system (Section 4.5), even though the model accurately represents interactions between variables (thus increasing the risk of false negatives). Therefore, in line with objective (2), Section 4.5 will demonstrate the inadequacy of the synchronous update mode in a case study and will propose using a non-synchronous alternative.

### 3.3. Deterministic or Not Deterministic?

Determinism is a property commonly found in historical (ecological) models [30,104]. In a deterministic model, initial conditions and the model structure are sufficient to predict the (unique possible) future. However, as knowledge about the system is often limited, including a certain amount of noise likely better reflects our ignorance and the intrinsic variability in interaction intensities. For instance, dispersion of individuals cannot be precisely predicted and is thus represented as single stochastic events or as a fixed dispersion rate representing the mean number of individuals leaving the system. Such stochastic events may have tremendous consequences on system dynamics and induce, e.g., priority effects [105] and alternative stable states [106,107]. Stochastic disturbances may also lead an ecosystem towards alternative thermodynamical trajectories [52]. Although determinism can be relevant for specific issues, objective (2) requires the explicit incorporation of uncertainty as random parameters, as a set of probabilistic transitions (e.g., Markov chains), or implicitly accounting for it (as in states-and-transition models).

### 3.4. Uncertainty as Stochasticity

Ecological dynamics are influenced by events (e.g., birth, storms, predation or dispersal) which can usually not be precisely identified and represented. In this case, a stochastic model can be convenient to account for such events. Stochastic models represent system dynamics as a set of probabilistic trajectories (sequences of discrete transitions). In particular, they can be used to predict the probability for a system to reach a given state after a given time or to take a given trajectory. Stochastic models compute the temporal evolution of a system based on its (past or current) state and the probability for a system to shift from one state to the other (called transition probability). This is typically what Markov chains do. Considering probabilistic changes is undeniably a more parsimonious approach and requires less information than a precise deterministic description. However, estimating transition probabilities [108] remains costly and more adapted to well-studied systems. Additionally, transition probabilities may also change over time, which requires even more information about the system. Therefore, when measurements are scarce, setting probabilities represents an assumption which can be avoided, thus not satisfying objective (2). Consequently, in Section 4, we will propose an alternative approach based on a non-probabilistic non-determinism called *possibilism* [15].

### 3.5. Predictive Capacity

Ecosystem dynamics can be represented through formal and non-formal models. Non-formal dynamical models such as succession diagrams [40], replacement sequences [91] or state-and-transition models [89] are well suited for ordering qualitative knowledge about the dynamics of a specific system. They are intuitive, which makes them practical, e.g., for decision support [96]. They represent all possible states and transitions according to observations, and thus they provide an exhaustive visualization of system dynamics, while explicitly representing ecological events. However, as these models are limited to observed ecosystem states, they are not predictive [100]. Moreover, their size is constrained by the fact that they are usually hand made and visually interpreted. Although they provide a valuable representation of available knowledge about system dynamics, their inability to infer ecosystem states with no historical analogue limits their applicability in a context of a changing environment due, for example, to climate forces or anthropogenic disturbances. As a consequence, they do not satisfy objective (4), and we will thus adopt a formal approach.

### 3.6. From Properties to an Innovative Formalism

All the aforementioned properties may be assembled in various ways (Figure 4). Modeling formalisms are thus composite objects that inherit the limitations of their constituents. As each specific assemblage is designed to address specific questions, we can now discuss the relevance of existing formalisms regarding the four objectives.

1.The ability to grasp the qualitative dynamics does not discard any formalism, as both quantitative and qualitative models can provide insights about the qualitative dynamics. However, loop analysis is restricted to equilibrium systems, which can hardly be known a priori and may be inappropriate for non-equilibrium systems commonly found in ecology and environmental sciences.2.As we aim to make as few assumptions as possible about parameters, we will discard quantitative formalisms for they impose strong constraints on data requirements (e.g., fixed or variable interaction coefficients, knowledge of functional forms) for building models. Estimating transition probabilities for Markov models also requires sufficient amounts of data, which are not always available.3.All formalisms can provide some form of explanation. However, some ecosystem models may act as black boxes and thus prevent a detailed and meaningful analysis. In contrast, models like state-and-transition models can enable tracking causal pathways leading to a particular outcome.4.Predictive capacity refers to the ability to forecast the future state of a system to some specific level of accuracy using a computational or mathematical model [100]. In this regard, non-formal models such as state-and-transition models are not predictive as they rely only on observed states and transitions between them and do not infer unobserved states.

Building on the vast literature and a long tradition in systems biology starting with [109,110], the aforementioned limitations of current formalisms led us to develop an innovative dynamical modeling framework in ecology called *ecological discrete-event networks* (EDEN) and freely available (https://github.com/fpom/ecco, accessed on 15 September 2023).

## 4. The Ecological Discrete-Event Networks (EDEN) Modeling Framework

### 4.1. A Brief Overview of the EDEN Framework

Before discussing the major properties of the EDEN framework in detail, we first illustrate them through a simple community model of species extinctions—or community disassembly—induced by trophic and competitive interactions ([111], submitted) (Figure 5). It is derived from experiments on protist communities in which basal species are fed with bacteria [107]. While ecological interactions (Figure 5a) are the driver of change, the model focuses on *qualitative changes* induced by these interactions (in accordance with objective (1)). In the EDEN framework, these qualitative changes are called *events*. These events are species extinctions (named by their initials) and are summarized as a set of if-then *rules* (Figure 5b). Starting from the {APT} state, the *asynchronous* execution (i.e., one by one) of these rules results in several alternative trajectories. Due to various factors (e.g., stochasticity, population sizes or interaction strength), these trajectories may have different probabilities of occurrence. However, the EDEN framework is *possibilistic* (i.e., non-probabilistic and non-deterministic) and thus aims to account for all possible alternative trajectories, irrespective of their probabilities and duration [15]. We will show that these three model properties (i.e., an event-based, asynchronous and possibilistic approach) contribute to satisfy objective (2). Computed dynamics are represented as a *state-transition graph* (STG, not to be confused with STMs, Section 2.2.4), representing cascading extinction events (Figure 5c). In this STG, nodes and edges represent states and transitions, respectively. The model predicts (objective (4)) three alternative trajectories with distinct extinction sequences, thus providing a historical explanation (objective (3)) for the two distinct stable states {T} and {∅}. The corresponding explanation is: “the extinction of *P* is a necessary but not sufficient condition for {T} to persist”. These results will be discussed in more detail in Section 4.4.

### 4.2. A Qualitative Perspective on Ecological Components

Ecosystems are continuously reshaping under the influence of internal and external factors. These changes do not manifest as jumps, but rather as more or less abrupt continuous variations. However, modeling such continuous phenomena often requires difficult parameter estimation, which is generally out of reach given technical and financial limitations. Biologists faced the same issue when they first aimed to model the dynamics of regulatory networks. They circumvented this limitation by abstracting gene expression to a switch-like behavior in which genes are either expressed (ON, 1) or not (OFF, 0) [112]. This marked the beginning of logical modeling in biology, the most famous example being Boolean networks [110]. Although this “logical caricature” [113] may seem excessively simple, it proved surprisingly insightful in theoretical (e.g., [110]) and applied cases [114]. In particular, it appeared to closely match the nonlinear nature of gene expression [115]. These simplified yet qualitatively valid and robust models enabled studying larger regulatory networks exhaustively without resorting to numerical simulations. However, the use of logical models did not percolate to ecology and to most environmental sciences. The first attempt was a Boolean predator–prey model proposed by [116] in 1979. He showed that the Boolean model displays the same cyclical behavior as the continuous model, while a slightly refined model displays the same stable states and cyclic attractor. More recently, a qualitative approach was adopted through the use of timed automata for addressing land use changes, coral reefs dynamics (e.g., [117]. This early attempt is promising for designing relevant abstractions of more complex quantitative models, yet with much lower data requirements. The qualitative approach in EDEN wagers on the decades-long use of qualitative models in systems biology, and it already provides valid and insightful approximations of the continuous behavior of ecological systems (objective (1)).

### 4.3. Discrete Events as the Basic Unit of Change

The EDEN modeling framework is not time-driven (i.e., no global clock is driving system changes) but event-driven, and thus belongs to discrete-event models [118]. In such models, changes happen at possibly irregular time intervals. A transition occurs when a variable crosses a threshold and thus becomes functionally present or absent, or functionally active/inactive. The functional presence of a variable corresponds to its ability to cause or prevent qualitative changes in other variables (Figure 5a). We will illustrate this phenomenon using a simple ecological process. In this case, a threshold delineates the range of values above which the variable (here soil moisture) becomes functional (induces seed germination). Hence, the emergence of seedlings resulting from seed germination is a discrete event, also called a *transition*. While the EDEN framework can also include multiple thresholds (multivalued framework, Figure 5b), we will mostly focus on the Boolean framework where only one threshold is considered, for the sake of simplicity. In the Boolean representation, when moisture is above (+) (resp. below, (−)) this “functionality threshold”, it is said to be “present” (resp. “absent”). The transition from a low to high germination rate is represented as an if-then rule expressing the following sentence: “If soil moisture is high enough (M+), then seedlings may develop (S+)”, in which the word “enough” indicates the moisture threshold below which germination cannot occur (here, the base water potential; [119]), and the word “may” indicates that the transition may not occur, even if soil moisture is sufficiently high (e.g., if the soil is below the base temperature; [119]). Formally, the system dynamics can be formalized with the rules:M+→S+
M−→S−

Note that Figure 6 implies that if soil moisture is insufficient, the reverse transition may occur. A rule consists of a left-hand side (called a condition, here M+) and a right-hand side (called a realization, here S+), with an arrow between them (noted →) representing the event (here, an effective germination). The condition as well as the realization of a rule can include one (as in the example) or several variables. Due to their high simplicity, such qualitative rules can be easily derived from any knowledge source (e.g., experts, observations or experimental data), while providing a highly general, yet realistic description of the transition considered. In the EDEN approach, we assume that such a functionality threshold exists for any interaction, such that the Boolean abstraction is always valid (i.e., captures the qualitative properties of the phenomenon of interest, [116]). The threshold value is unknown a priori (Figure 6a). This is an advantage as it allows building a model with highly limited knowledge about the system under study. Indeed, ecological threshold values are costly to measure and are often highly variable in time and space, thus hampering any threshold estimation for most ecological interactions [120,121].

However, variables are not equally sensitive to a given input. Consider now two plant species (with their respective seedlings $S_{1}$ and $S_{2}$) with specific soil moisture requirements for germination. If one seeks a more detailed representation, we can substitute the previous model for a multivalued one. In this case, we use four different rules associated with three different soil moisture thresholds (0, 1 and 2) (Figure 6b):$$M\geq 1\to S_{1}+$$
$$M\geq 2\to S_{2}+$$
$$M<1\to S_{1}-$$
$$M<2\to S_{2}-$$

In the Boolean framework, as we ignore quantitative differences, if $S_{1}$ (i.e., the species with the lowest moisture requirement) can germinate, then $S_{2}$ can too (Figure 6b). Thus, we have:$$M+\to S_{1}+$$
$$M+\to S_{2}+$$
$$M-\to S_{1}-$$
$$M-\to S_{2}-$$

However, the way rules are executed is not yet specified. When moisture is sufficient (M+), are rules executed simultaneously or separately? This choice is likely to strongly impact the modeled dynamics, and should thus be carefully considered. In the example, if rules are executed simultaneously (i.e., synchronously), the Boolean description can miss important trajectories. For instance, soil moisture may be sufficient for $S_{1}$ and $S_{2}$ to germinate, but hidden variables such as temperature may prevent or delay the germination of one species, thus inducing a non-synchronous response between $S_{1}$ and $S_{2}$. So, how can we represent all qualitatively realistic trajectories, accounting for the effect of hidden variables, while avoiding adding quantitative information or other variables?

### 4.4. Accounting for Uncertainty in the Event Timing: The Asynchronous Update Mode

In a given state of an ecological system, several events may occur (for instance, when soil is moist, several plants may germinate). How does the model manage these concurrent events? This question is related to the way variables change at each computational step, called the *update mode* [122]. As mentioned in Section 3.2, variables can be updated synchronously, which is the case in most ecological models. It is justified in a continuous time perspective where process rates can be adjusted by parameters (e.g., in differential equations), but becomes problematic in a qualitative perspective. We illustrate here the limits of the synchronous update mode using the APT model discussed in Section 4. When rules of the APT model (Figure 5b) are executed synchronously (Figure 7a), the dynamics are deterministic and the {APT} state reaches only one stable state (∅). However another end state, {T}, was observed experimentally (see Appendix 1 in [107]). In this case, either the model structure (i.e., interactions between variables) is wrong, or the way events are scheduled is inappropriate for the studied phenomenon.

We test the second hypothesis by relaxing all assumptions about the timing of extinctions (determined by interaction strength and population sizes, which are generally uncertain), and execute rules asynchronously, i.e., one by one (Figure 5c and Figure 7b). As each rule changes the state of only one variable, this is similar to the fully asynchronous mode used in Boolean networks [109]. In this asynchronous update mode, transitions are not driven by a global clock synchronizing them, but rather have their own timing. The asynchronous model not only predicts two stable states and four transient states, which are all observed experimentally, but also all observed transitions ([111], submitted). Note, however, that some synchronous Boolean networks also proved insightful for some ecological phenomena (e.g., [14]). Additionally, synchronizing specific events can be relevant as it may more closely match available knowledge. Therefore, in order to make the formalism more flexible for users and in accordance with objective (2), we adopted a *partially synchronous* update mode, in which a rule can update several variables simultaneously (not shown here, but see [15]), while rules are still executed one by one. This is similar to semi-synchronic Boolean networks used in [123].

### 4.5. Possibilism as an Innovative Approach to Non-Determinism

Assuming that ecological systems are non-deterministic, the trajectory they take depends on several factors, such as event timing and interaction strengths. Generally, only a few alternative trajectories will be frequently observed, thus motivating a probabilistic approach. However, rare events critically contribute to history in ecological and biological dynamics [124,125], thus making the exhaustive set of trajectories highly relevant, whatever their probabilities [126,127,128]. Therefore, we adopt a *possibilistic* perspective, in which all the possible trajectories are computed, that is, all changes compatible with the predefined variables and rules, regardless of probabilities. This perspective combines with the asynchronous update mode, which usually opens several alternative trajectories (as in Figure 7b). It allows assessing all the far-reaching consequences of the occurrence or non-occurrence of a given event (e.g., management action or natural event). Possibilism has been widely used in systems biology to study the transient and asymptotic behaviors of regulatory networks in response to various environmental stimuli [129]. It has also been used to disentangle the complex sequences of events leading to a particular outcome, namely a causality analysis, based on counterfactual reasoning [130], which may be useful for ecosystem management and decision support [131]. Note, however, that possibilism is only relevant for a coarse-grain system description. Indeed, a possibilistic description of the behavior of each individual in a population would lead to a huge and inextricable set of trajectories that would provide little insights, if any.

Following our review of ecosystem model types (Section 2), we developed the EDEN framework to achieve our stated objectives (Section 3) by combining the model properties we presented in this section. The qualitative approach implies that only major changes will be considered, while the discrete-event framework implies that these changes may eventually occur at irregular time intervals, following the various threshold crossings. Additionally, the asynchronous occurrences of events open alternative trajectories corresponding to contrasted event sequences, which are computed in a possibilistic way (i.e., exhaustively). The EDEN framework has already succeeded in representing and modeling realistic social-ecological systems and their dynamics in specific case studies. EDEN models have already helped in understanding terrestrial temperate [132], tropical [133,134] and aquatic ([111], submitted) ecosystems, either for applied [133,134] or theoretical objectives [15]. In each of these case studies, the ecosystem is represented as an interaction network, then handled by the EDEN rules formalizing the way this network could change over time (see pre-mentioned references for detailed modeling steps).

### 4.6. The State-Transition Graph as the Assemblage of Model Properties

As in vegetation succession diagrams [83], state-and-transition models [89,90], qualitative reasoning [78], community assembly models [106,135,136] or timed-automata [137], the EDEN framework represents ecological dynamics as a state-transition graph (STG).

The topology of this graph provides valuable information about ecological dynamics [15,106,136,138]. An STG generally includes three main topological structures (also called graph components), namely, strongly connected components, stable states and basins [118,133]:A *strongly connected component* (SCC) is a set of mutually reachable states, i.e., any system change in it is reversible. It can be cyclic (only one trajectory, e.g., yellow states, Figure 8a) or complex (several trajectories, e.g., green states, Figure 8a), highlighting the presence of one or several feedback loops, respectively. Cyclic SCCs are discrete analogues of limit cycles [139] and have been used in community assembly to define cyclic changes in species community composition [106,136]. On the other hand, complex SCCs have been observed in cell differentiation [138], rangeland dynamics [90,140] and geomorphology [141], and have been predicted theoretically [118]. In addition, this concept is crucial in state-and-transition models for defining the concept of “state”, which is “a sustained equilibrium that is expressed by a specific suite of vegetative communities” [90].A *stable state* or deadlock is a single state with no successor (Figure 8, dark blue state). These have been interpreted as final phenotypes in cell differentiation trajectories [138] or non-invadable communities in community assembly experiments [142].Finally, *basins* are defined as sets of states which (1) are not part of an SCC or stable state and (2) all lead to the same SCCs or stable states (Figure 8, orange and non-terminal blue states). Although they do not have well-known empirical counterparts, a recent model based on protist community disassembly experiments [107] confirmed the relevance of such structures, suggesting their role as sets of transient states with indeterminate fate.

There is no mathematical limit to the size of STGs; EDEN model STGs sometimes grow up to thousands or billions of states (e.g., [132,133]). Automatic analyses (summarized in [143]) are thus required to identify relevant information. They include (but are not restricted to) STG aggregations and model-checking [134,144]. STG aggregation is related to the identification of transient and persistent sets of states and consists in merging states according to their neighboring relations [145,146]. On the other hand, model-checking refers to the formal verification of dynamical properties in the set of computed system trajectories. It is beyond the scope of this paper to provide an exhaustive list of such techniques, so we will briefly survey the main analysis techniques.

The main STG aggregation aims to summarize the STG by merging its topological components (i.e., SCCs, stable states and basins), thus forming a *hierarchical transition graph* ([145], Figure 8b). In this aggregated graph, we observe a transition between two components (for example, between green and blue components, Figure 7b) only if there is a transition between at least two states belonging to either of these components (Figure 8a). Nodes of the hierarchical transition graph correspond to transient (basins) and persistent behaviors (SCCs and stable states), while transitions represent the irreversible changes. Note that the similarity between the definition of SCCs and “states” in the state-and-transition modeling framework make the hierarchical transition graph comparable with a state-and-transition model.

On the other hand, model-checking is a verification method which automatically proves if a given state-transition graph satisfies a dynamical property, usually expressed as a temporal logic formula [144]. However, the mathematical formalism used for expressing dynamical properties is usually difficult to master for ecologists and biologists. Therefore, pre-established properties (called query patterns) already exist in biology [129] and ecology [117,134]. These query patterns enable asking questions such as:Is an ecosystem collapse avoidable?Is a productive ecosystem state reachable and stable (e.g., included in an SCC)?Is this productive state always preceded at some time by, say, a disturbance or a specific process (rule)?

While temporal logic has mostly been used in systems biology, seminal applications can be found in agricultural sciences [147,148] or ecosystem management [117,134].

## 5. Discussion and Conclusions

All (social-) ecological systems are complex systems requiring inter- and often multi-disciplinary approaches, which modeling may undeniably help understanding and predicting. Dynamical ecosystem models aim at predicting temporal changes in ecosystem state variable values induced by internal system structure and/or external influences. So far, ecosystem modeling has mostly adopted quantitative methods, often based on differential equations. However, depending on the objectives, quantitative information may be unnecessary, e.g., the search for keystone species [74], study of vegetation successions [149] or qualitative response of ecological communities to press perturbations [150]. Additionally, parameter values are often imprecisely known and may change over time, thus reducing confidence in some model results.

In this regard, qualitative models represent an alternative in data-poor situations. Although less precise, they are no less rigorous and often rely on fewer assumptions, thus increasing the generality of model predictions. They are not meant to replace quantitative models in all situations, but can prove useful in the early steps of the modeling process for generating and rejecting qualitative hypotheses. As they sometimes display ambiguous predictions, they can also inform ecologists about which processes require quantitative information to provide unambiguous qualitative predictions and which do not. Even in data-rich situations, qualitative models may provide a relevant alternative for understanding the (social-ecological) system functioning, for example, by revealing its long-term dynamics [133,134]. In such a context, qualitative models may also provide intuitive and user-friendly tools, as they may formalize expert knowledge and everyday narratives.

We have listed some of the major properties of each modeling method, illustrated in complex ecosystem ecology, and discussed their limitations according to some specific objectives. We expected ecosystem models to (1) grasp the qualitative dynamics of the system of interest, (2) make as few assumptions as possible about interaction parameters and spatial structures, (3) be explanatory and (4) predictive. We have shown that there is a long tradition of qualitative approaches in ecology, dating back to [63], followed by [11,40,89], to name a few. In particular, after others (e.g., [14,147]), we propose the EDEN modeling approach, which considers discrete events as the basic units of change in qualitative dynamics. An event is a change in system state defined as the threshold crossing which delimits different ecological functions of a variable.

We also highlighted some of the assumptions implied by a deterministic view (at least) in qualitative models. As illustrated in Figure 7 from the rule set in Figure 5b, synchronous qualitative models assume equally fast processes, which often is a too strong assumption and can lead to the rejection of a model that would be accurate if asynchronous. As a consequence, we suggest considering the asynchronous update mode as a realistic alternative for qualitative models. Note, however, that determinism still enables the prediction of important phenomena such as trophic cascades [63]. Probabilistic models also face limitations due to necessary assumptions. Indeed, a probability distribution needs to be chosen for any stochastic process, and every choice of distribution needs to be justified, thus requiring much information. When such information is unavailable or insufficient, we propose possibilistic non-determinism as a valuable alternative. It infers all possible alternative trajectories given a set of premises (rules or equations). However, the number of possible computed states grows exponentially with the number of involved variables or variable values, thus making model computation, analysis and intelligibility challenging. Therefore, possibilism does not prevent modelers from keeping models simple. Although we already applied the EDEN framework to some concrete needs, a remaining issue is to demonstrate its relevance to most environmental sciences.

We think qualitative models raise important methodological questions in ecological modeling. In particular, the prevalence of quantitative models, in spite of the poor quantitative information, suggests a confusion between precision and accuracy. For instance, the APT model (Figure 5) is imprecise (i.e., no numerical values of population size are given), but is simultaneously highly accurate since all predicted states and transitions are actually observed [107]. This does not discard quantitative models as relevant tools for explaining complementary aspects of ecological dynamics. It is common to find aspects of ecological phenomena whose explanation requires the use of multiple models (what [151] calls model pluralism). Model pluralism is a fact in ecology and should be maintained and promoted. By offering various mathematical expressions to ecological situations, it widens the range of ecological questions one can ask to a particular system and thus helps formulating and testing new ideas. Another remaining issue in this regard is to adequately couple quantitative and qualitative models, in a way exploiting the most relevant properties of each framework for addressing specific scientific questions.

In this respect, systems biology can be a great source of inspiration for ecology. Its constant dialog with computer sciences contributed to its current spectrum of qualitative-to-quantitative formalisms, from the Boolean to multivalued, hybrid and fully continuous methods [152,153]. There is a continuous effort to bridge gaps between (and not necessarily unify) existing methods, which is of great interest for explaining natural phenomena. As Levins [11] puts it, talking about qualitative models: “general models are necessary but not sufficient for understanding nature. For understanding is not achieved by generality alone, but by a relation between the general and the particular”. Mathematical connections between, on the one hand, differential equations and, on the other hand, loop analysis [69], logical models [154,155] or qualitative reasoning models [79] have already been demonstrated. Therefore, if each qualitative model is proved to be a relevant abstraction of specific aspects of quantitative models, it is possible to draw relevant conclusions with much less information. This is encouraging as modelers can lean on this pluralism for building more robust explanations of natural phenomena.

## Figures and Tables

**Figure 3 entropy-25-01526-f003:**
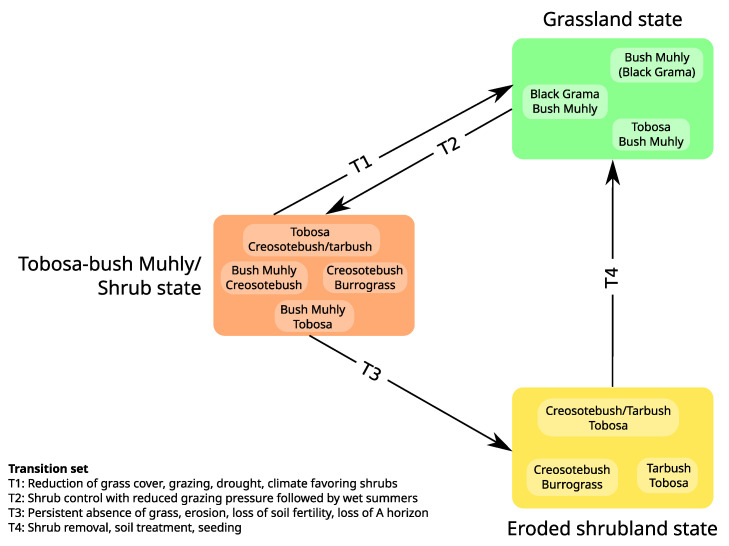
**State-and-Transition Model**. Boxes inside each state correspond to community phases linked through community shifts. While community pathways may not be explicitly represented (as is the case here), community phases within one state should always be mutually reachable. Additionally, a shift between two community phases is generally considered bidirectional. Transitions (arrows) are considered irreversible unless substantial management efforts are engaged. Adapted from [95].

**Figure 4 entropy-25-01526-f004:**
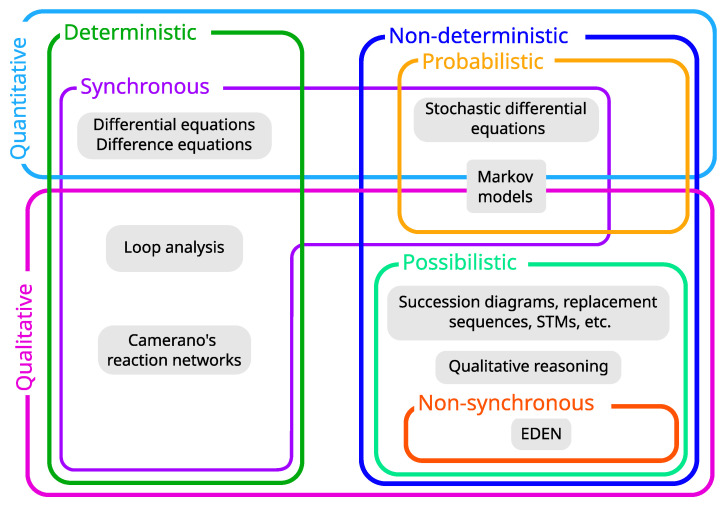
**Assemblages of properties of ecosystem dynamical models**. Note that some model types may have undefined/unapplicable properties. For instance, despite their non-deterministic nature, the properties ”synchronous" or ”non-synchronous” are not rigorously applicable to STMs.

**Figure 5 entropy-25-01526-f005:**
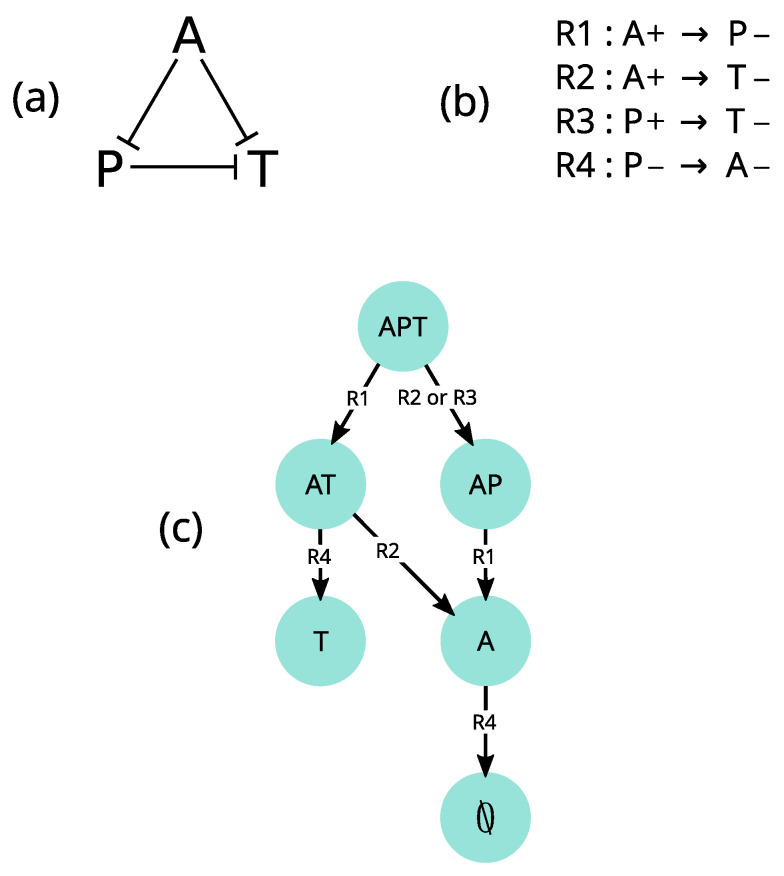
**Discrete-event model of community disassembly**. (**a**): Interaction graph of three protist species: *Amoeba proteus* (*A*), *Paramecium caudatum* (*P*) and *Tetrahymena pyriformis* (*T*); edges represent directed negative interactions. (**b**): Rule set describing species extinctions and their conditions. These extinctions result from the following interactions: *A* eats and depends on *P*; *A* eats *T*; and *P* competes (unidirectionally) with *T*. (**c**): The state-transition graph resulting from all possible rule executions from the initial state {APT}. State (nodes) labels indicate which species are present, while transition (edges) labels indicate which rules can be executed for each transition. Note that one transition can result from several alternative rule executions (as in the transition from {APT} to {AP}).

**Figure 6 entropy-25-01526-f006:**
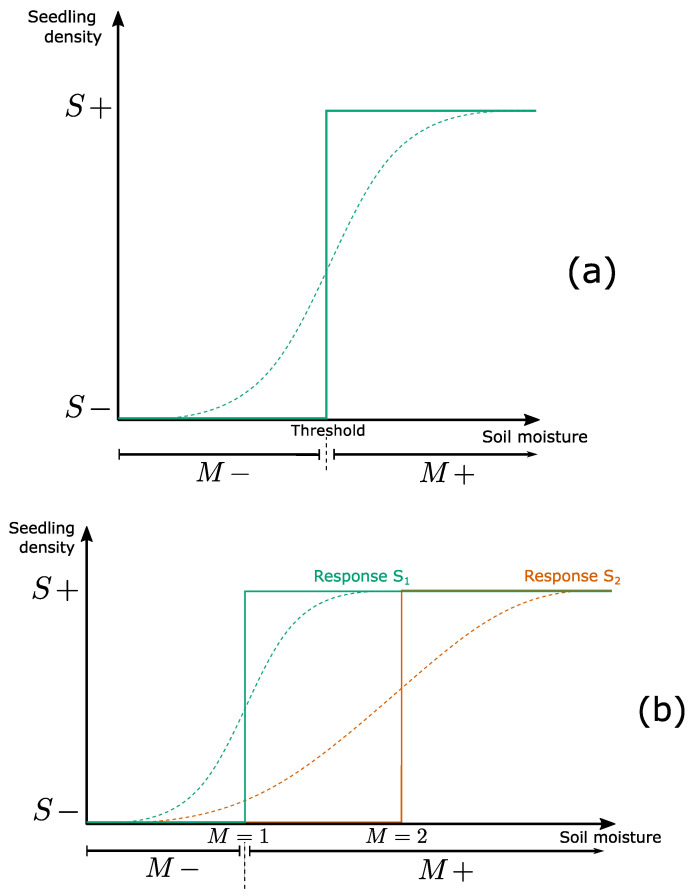
**Threshold-based quantization of continuous dynamics.** (**a**): Curves representing the interaction between soil moisture *M* and seedling density *S* (and corresponding to the rules M+→S+ and M−→S−). The sigmoid curve (dotted line) illustrates one possible monotonically increasing function linking the two variables. The continuous line corresponds to its step-function (Boolean) approximation. For the sake of representation, the threshold between M− and M+ is clearly positioned but may be more fuzzy in reality. (**b**): Comparison between the Boolean and multivalued representations of interactions. Sigmoid curves represent the effect of soil moisture on the germination of two species $S_{1}$ and $S_{2}$. In the Boolean abstraction (+ and − intervals on x-axis), there is no difference between specific thresholds. Therefore, if moisture is sufficient for one species, then it is sufficient for all. Inspired from [109].

**Figure 7 entropy-25-01526-f007:**
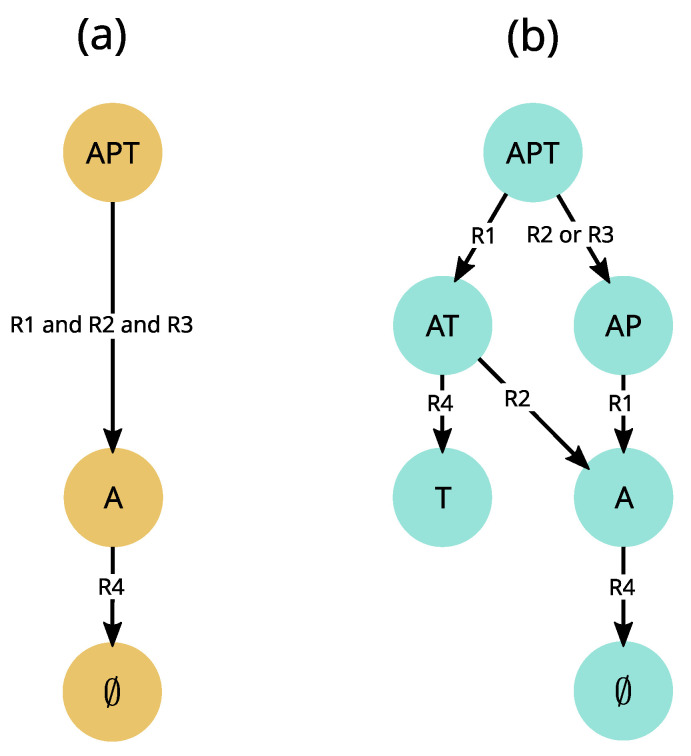
**Comparison of trajectories generated by synchronous and asynchronous update modes derived from the APT model described in **Figure 6. (**a**) In the synchronous update mode, all valid transitions are fired simultaneously. For instance, in the {APT} state, rules R1, R2 and R3 are satisfied (see Figure 6b for rule definitions) and are thus executed simultaneously. The resulting dynamics are thus deterministic and necessarily end up in the empty state ∅. (**b**) In contrast, the asynchronous update mode used in EDEN opens an alternative trajectory from the {APT} state by separately firing, on the one hand R2 or R3, and on the other hand R1, possibly leading to the alternative stable state {T} observed in [107] experiments.

**Figure 8 entropy-25-01526-f008:**
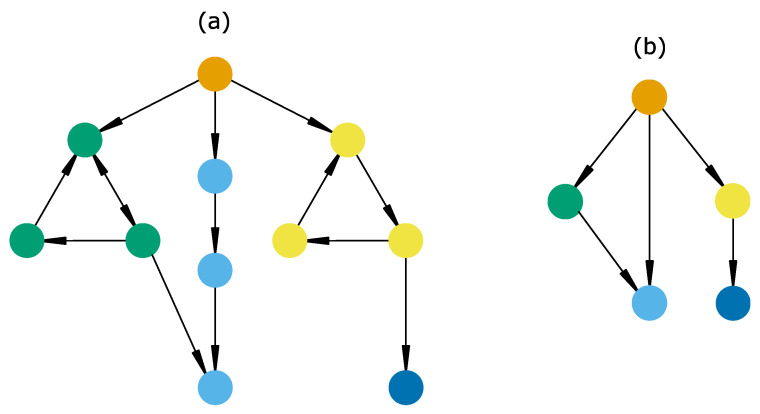
**Illustration of topological structures in a state-transition graph and its compression in a hierarchical transition graph.** (**a**) State-transition graph (STG). Each node and edge is a state and a transition, respectively. Node colors indicate which topological structure a state belongs to. For instance, all yellow states belong to the same SCC. Note that the light blue states represent a set of states necessarily ending in a stable state and thus represent its (strong) basin. (**b**) Hierarchical transition graph corresponding to the STG in (**a**). Node colors thus match those of (**a**). Each node is a topological structure of the STG, and each transition is necessarily irreversible.

## Data Availability

Not applicable.

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
