# Peer review of "On the History of Ecosystem Dynamical Modeling: The Rise and Promises of Qualitative Models"

_entropy, 2023, doi:10.3390/e25111526_

Round 1

Reviewer 1 Report

Comments and Suggestions for Authors

This is a well written and informative paper. It is outside my core area of expertise so I found it a ‘good read’ providing background and history of the development of the field.

Specific comments.

#101 – might also refer to “biological plasticity” organisms change their behaviour (parameter values) in response to changing circumstances.

Somewhere, possibly in section 1.1, probably worth mentioning the issue of feedbacks and sensitive dependence on initial conditions in these models based on multiple, non-linear, differential equations.

#275 – ‘expose’ word not needed. Who is the “our”, I don’t think it is just the authors of this paper, is it the ecological community? 

#276 The syntax does not work. “ We need to:

1.     Grasp…

2.     Make… 

3.     Be…

4.     Be

#557 ?? – update

The “Discussion and Conclusion” section is a restatement of the main arguments and does not really address the authors views in context of the literature. As a review this has been done to a large extent, but a focus on remaining issues and the way forward would be useful.

Comments on the Quality of English Language

The English is generally very good but there are some very odd word choices in places.

Author Response

This is a well written and informative paper. It is outside my core area of expertise so I found it a ‘good read’ providing background and history of the development of the field.

Thank you for this positive appreciation.

Specific comments.

#101 – might also refer to “biological plasticity” organisms change their behaviour (parameter values) in response to changing circumstances.

Thank you, done. 

Somewhere, possibly in section 1.1, probably worth mentioning the issue of feedbacks and sensitive dependence on initial conditions in these models based on multiple, non-linear, differential equations.

Good point, we added a few words about it.

#275 – ‘expose’ word not needed. Who is the “our”, I don’t think it is just the authors of this paper, is it the ecological community?

Both: this sentence mentioned the objectives of the present study (and that of the authors, indeed), but we also insisted that these objectives concern the ecological community.

#276 The syntax does not work. “We need to:

  1. Grasp…
  2. Make…
  3. Be…
  4. Be…

#557 ?? – update

Thank you, it is now corrected.

The “Discussion and Conclusion” section is a restatement of the main arguments and does not really address the authors views in context of the literature. As a review this has been done to a large extent, but a focus on remaining issues and the way forward would be useful.

Indeed, the conclusion is a summary of the main argument of our paper. Yet, as we cannot rewrite the whole discussion, we added one sentence about the main remaining issue related to the EDEN framework (i.e., its applicability).

Comments on the Quality of English Language

The English is generally very good but there are some very odd word choices in places.

Thank you, we edited the whole document.

Submission Date

21 September 2023

Date of this review

10 Oct 2023 00:45:40

Reviewer 2 Report

Comments and Suggestions for Authors

Dear authors,

Thank you for the opportunity to read this interesting paper. Please find my comments for improvement below section by section:

Introduction: The introduction appears to be well-structured and provides a clear overview of the paper's content. It starts with general background information and then gradually narrows down to the specific focus of the paper. The mention of the "computational turn" in the 1950-60s is a relevant point, but it could benefit from a bit more elaboration to explain how this turn is connected to ecosystem modelling. It might be useful to briefly describe how computer simulations and complex models became integral to ecosystem research during this period. On the other hand, despite the introduction effectively introduces the distinction between quantitative and qualitative models and explains the limitations of quantitative models in data-poor situations, it might be helpful to provide a concrete example to illustrate this point.

The introduction mentions the development of various qualitative modelling frameworks since the 1980s. I believe that it might be beneficial to briefly explain the key characteristics or principles of some of these frameworks to give readers a sense of the diversity within qualitative modelling. Also, when it comes to the proposal of the Ecological Discrete-Event Networks (EDEN) framework, it might be useful to provide a sentence or two explaining what EDEN is or how it differs from existing qualitative frameworks at this stage.

Overview of ecosystem ecology and dynamical modelling approaches: The introduction of thermodynamical principles in ecology is intriguing and adds a unique perspective. However, it could benefit from a bit more explanation for readers not familiar with this approach.

Ecosystem models’ properties and their limitations: While the passage provides a theoretical overview of various modelling properties, incorporating concrete examples or case studies could enhance comprehension. Concerning the limitations associated with certain modeling approaches, I consider that a more in-depth exploration of these limitations would be useful to give robustness to your approach. Providing specific examples of scenarios where these limitations become apparent would be insightful. Since the passage introduces EDEN as an innovative formalism, it would be helpful to provide a brief overview of how EDEN addresses the limitations discussed earlier.

The Ecological Discrete Event Networks (EDEN) modelling framework: Expanding on how the EDEN framework has been applied in ecological research or practical scenarios would provide readers with a deeper understanding of its real-world utility. Including case studies or examples of ecological problems solved using EDEN would be informative.

When introducing model-checking as a verification method, it might be helpful to provide a brief explanation or reference for readers who are less familiar with this concept.

Lastly, since the section focuses on the strengths and advantages of the EDEN framework, it would be beneficial to acknowledge any potential limitations or challenges associated with its implementation in ecological modelling.

Good luck with your revision!

Comments on the Quality of English Language

Minor revision is needed

Author Response

Dear authors,

Thank you for the opportunity to read this interesting paper. Please find my comments for improvement below section by section:

Introduction: The introduction appears to be well-structured and provides a clear overview of the paper's content. It starts with general background information and then gradually narrows down to the specific focus of the paper. The mention of the "computational turn" in the 1950-60s is a relevant point, but it could benefit from a bit more elaboration to explain how this turn is connected to ecosystem modelling. It might be useful to briefly describe how computer simulations and complex models became integral to ecosystem research during this period.

Yes, good point, although we are not historians and are not able to elaborate too much in this direction. We added a few sentences here.  

On the other hand, despite the introduction effectively introduces the distinction between quantitative and qualitative models and explains the limitations of quantitative models in data-poor situations, it might be helpful to provide a concrete example to illustrate this point.

Indeed, it is useful. We present an illustration based on our past studies in dendrochronology (in which a quantitative model parametrized on more than ten parameters was showing instability and equifinality).

The introduction mentions the development of various qualitative modelling frameworks since the 1980s. I believe that it might be beneficial to briefly explain the key characteristics or principles of some of these frameworks to give readers a sense of the diversity within qualitative modelling. Also, when it comes to the proposal of the Ecological Discrete-Event Networks (EDEN) framework, it might be useful to provide a sentence or two explaining what EDEN is or how it differs from existing qualitative frameworks at this stage.

Actually, the qualitative framework principles are given in more details right in the following section (1). So, we did not add any sentence here. Also, we already tried to list further (section 3.6) most EDEN specificities. Yet, we added here one single sentence insisting that EDEN intends to simultaneously combine the four above-mentioned objectives.  

Overview of ecosystem ecology and dynamical modelling approaches: The introduction of thermodynamical principles in ecology is intriguing and adds a unique perspective. However, it could benefit from a bit more explanation for readers not familiar with this approach.

OK, we added some explanations in this direction, although it would not be the place to provide a detailed survey of the “extremal principle” attempts.

Ecosystem models’ properties and their limitations: While the passage provides a theoretical overview of various modelling properties, incorporating concrete examples or case studies could enhance comprehension. Concerning the limitations associated with certain modeling approaches, I consider that a more in-depth exploration of these limitations would be useful to give robustness to your approach. Providing specific examples of scenarios where these limitations become apparent would be insightful. Since the passage introduces EDEN as an innovative formalism, it would be helpful to provide a brief overview of how EDEN addresses the limitations discussed earlier.

Yes, but it is always difficult and uncomfortable to mention studies showing clear limits and flaws. Hence, we focused again on the (our) dendrochronological illustration (above-mentioned).  

Concerning how EDEN addresses the limitations previously mentioned, we really think that the asynchronous update mode and the qualitative properties combine to circumvent the present limitations most ecological models are experiencing today. Sentences added. 

The Ecological Discrete Event Networks (EDEN) modelling framework: Expanding on how the EDEN framework has been applied in ecological research or practical scenarios would provide readers with a deeper understanding of its real-world utility. Including case studies or examples of ecological problems solved using EDEN would be informative.

Ok, we added and cited some of the recent EDEN applications that are real successes of this formalism. We mentioned some of their specificities. (No need in this review paper to go deeper into the details of such published studies.)

When introducing model-checking as a verification method, it might be helpful to provide a brief explanation or reference for readers who are less familiar with this concept.

This has already been done. We are able to briefly describe the method and to exhibit some of its recent successes. Now added …

Lastly, since the section focuses on the strengths and advantages of the EDEN framework, it would be beneficial to acknowledge any potential limitations or challenges associated with its implementation in ecological modelling.

Good luck with your revision!

Sincerely  

Comments on the Quality of English Language

Minor revision is needed

Submission Date

21 September 2023

Date of this review

29 Sep 2023 16:32:37
